# Cobalt (II) Chloride Regulates the Invasion and Survival of *Brucella abortus* 544 in RAW 264.7 Cells and B6 Mice

**DOI:** 10.3390/pathogens11050596

**Published:** 2022-05-18

**Authors:** Tran X. N. Huy, Trang T. Nguyen, Alisha W. B. Reyes, Heejin Kim, WonGi Min, Hu J. Lee, John H. Lee, Suk Kim

**Affiliations:** 1Institute of Applied Sciences, HUTECH University, 475A Dien Bien Phu St., Ward 25, Binh Thanh District, Ho Chi Minh City 72300, Vietnam; txn.huy@hutech.edu.vn; 2Institute of Animal Medicine, College of Veterinary Medicine, Gyeongsang National University, Jinju 52828, Korea; nguyentrang2907@gmail.com (T.T.N.); khjin0704@gnu.ac.kr (H.K.); wongimin@gnu.ac.kr (W.M.); hujang@gnu.ac.kr (H.J.L.); 3Department of Veterinary Paraclinical Sciences, College of Veterinary Medicine, University of the Philippines Los Baños, College, Laguna 4031, Philippines; awbreyes@gmail.com; 4College of Veterinary Medicine, Chonbuk National University, Iksan 54596, Korea; johnhlee@chonbuk.ac.kr

**Keywords:** *Brucella abortus*, CoCl_2_, HIF-1α, immune response, RAW 264.7 cell, B6 mouse

## Abstract

The effects of Cobalt (II) chloride (CoCl_2_) in the context of *Brucella abortus* (*B. abortus*) infection have not been evaluated so far. Firstly, we found that CoCl_2_ treatment inhibited the phagocytosis of *B. abortus* into RAW 264.7 cells. The inhibition of bacterial invasion was regulated by F-actin formation and associated with a reduction in the phosphorylation of ERK1/2 and HIF-1α expression. Secondly, the activation of trafficking regulators *LAMP1*, *LAMP2*, and lysosomal enzyme *GLA* at the transcriptional level activated immune responses, weakening the *B. abortus* growth at 4 h post-infection (pi). The silencing of HIF-1α increased bacterial survival at 24 h pi. At the same time, CoCl_2_ treatment showed a significant increase in the transcripts of lysosomal enzyme *HEXB* and cytokine *TNF-α* and an attenuation of the bacterial survival. Moreover, the enhancement at the protein level of HIF-1α was induced in the CoCl_2_ treatment at both 4 and 24 h pi. Finally, our results demonstrated that CoCl_2_ administration induced the production of serum cytokines IFN-γ and IL-6, which is accompanied by dampened *Brucella* proliferation in the spleen and liver of treated mice, and reduced the splenomegaly and hepatomegaly. Altogether, CoCl_2_ treatment contributed to host resistance against *B. abortus* infection with immunomodulatory effects.

## 1. Introduction

*Brucella abortus* (*B. abortus*) is one of the twelve currently recognized *Brucella* species and causes disease in cattle. Additionally, the other *Brucella* species also have different preferential host specificities, including *B. melitensis* (goat and sheep); *B. suis* (pig, hare, reindeer, caribou, and rodent); *B. ovis* (sheep); *B. neotomae* (desert rat); *B. canis* (dog); *B. ceti* (dolphins); *B. pinnipedialis* (seals); *B. microti* (wild voles); *B. inopinata* (human); *B. papionis* (baboons); and *B. vulpis* (red foxes) [1]. They are the causative agents of brucellosis, a common zoonosis in the world. Among them, *B. inopinata*, *B. abortus*, *B. suis*, and *B. melitensis* are the most pathogenic species in humans. Both human and animal brucellosis have caused negatively severe medical and economic impacts [2,3]. These impacts are due to the stealthy strategies of *Brucella* to provoke a systemic infection, resulting in the dissemination of bacteria to different organs, tissues, and cells. Moreover, the most crucial virulence determinant, the Type IV secretion system (T4SS), facilitates the successful evasion of the host bactericidal activity and the establishment of an intracellular replicative niche, leading to chronic infection. For these reasons, the diagnosis, prevention, and treatment of brucellosis have become complicated and challenging [4,5].

Cobalt is well-known as a core element of vitamin B12 and an essential trace element in the human diet. It plays a crucial role in forming enzymes such as methionine aminopeptidase in bacteria and yeast and nitrile hydratase in actinomycetes and bacteria [6,7]. In addition to causing harmful impacts such as lipemia, cardiomyopathy, and hard metal asthma of long-term exposure to a high concentration, Cobalt (II) chloride (CoCl_2_) can display advantageous effects in anemia treatment, fracture healing in mice, and anti-*Helicobacter pylori* (*H. pylori*) activity [8,9,10]. Recently, many studies have reported the stabilization of transcription factor HIF-1α induced by CoCl_2_ by a proven mechanism to mimic hypoxia. Notably, HIF-1α functions as a central regulator of oxygen and energy homeostasis, cellular metabolism, and host immunity [11,12]. A study by Zenk et al. [13] demonstrated that a stabilized HIF-1α decreased the intracellular growth of *Mycobacterium tuberculosis*. Furthermore, another study by Hwang et al. [14] showed an enhancement of antiviral activity associated with CoCl_2_-stabilized HIF-1α in a T98G cell line.

Briefly, the prevention and treatment of brucellosis have been based on vaccines and antibiotics. Herein, the antibiotic therapy is often empiric and relies on the choice of treatment regimen and duration of various antibiotics such as doxycycline, aminoglycosides, rifampicin, minocycline, trimethoprim, sulfamethoxazole, and quinolone [15]. *Brucella* are intracellular bacteria and reside in intracellular compartments that require cell-penetrating antibiotics to eliminate. However, this kind of antibiotic remains challenging, including retention and accumulation in host cells and penetration into bacteria-containing compartments [16]. Additionally, although live attenuated vaccine strains, including *B. abortus* S19, *B. abortus* RB51, and *B. melitensis* Rev.1, are effective against brucellosis in cattle and small ruminants, they still have some drawbacks [17]. Recently, immunotherapy has emerged as a potential alternative for bacterial infection treatment, especially for intracellular bacteria [18]. Delving deeper into understanding the defensive mechanisms of host immunity and cellular effectors is a promising approach in immunotherapy. Recently, CoCl_2_ has emerged as a promising therapeutic agent for the prevention and treatment of various cancers, as well as inflammatory diseases [19,20,21]. In this study, we carried out in vitro and in vivo experiments in RAW 264.7 cells and B6 mice, respectively, using CoCl_2_ as a stimulator of the host immune responses against *B. abortus* infection.

## 2. Results

### 2.1. CoCl_2_ Treatment Inhibited the Invasion of B. abortus into RAW 264.7 Macrophage Cells and Affected Brucella survival in Axenic Culture

The 3-(4,5-dimethylthiazol-2yl)-2,5-diphenyl-2H-tetrazolium bromide (MTT) assay was performed to evaluate the cytotoxic effect of CoCl_2_ on RAW 264.7 cells. The 10 mg/mL, 1 mg/mL, and 100 µg/mL concentrations of CoCl_2_ significantly induced cytotoxicity in the cells, whereas all lower concentrations had no cytotoxic effect compared to the control cells (Figure 1A). Therefore, the highest noncytotoxic concentration of CoCl_2_ (10 µg/mL) was used for the subsequent experiments. The result of the bacterial internalization assay showed that, at the early stage of infection, the number of bacteria in the CoCl_2_-treated cells was lower than in the control cells but not significant. However, at both 30 and 45 min after infection, this reduction became significant (Figure 1B). At the same time, the immunoblotting analysis demonstrated that CoCl_2_ treatment led to a decrease in the F-actin formation and phosphorylation of ERK1/2 in *B. abortus*-infected cells compared to the nontreated infected control cells at 30 min post-infection (pi). Notably, CoCl_2_ treatment induced a marked expression of HIF-1α compared to the untreated cells. In addition, *Brucella* infection reduced the HIF-1α expression in CoCl_2_-treated cells compared to uninfected cells (Figure 1C,D). Moreover, in an axenic culture condition containing sufficient nutrients to maintain *Brucella* viability during the experimental period, the survival rates of *Brucella* decreased significantly at all concentrations of CoCl_2_ at 24 h and in higher concentrations (10, 100, and 1000 µg/mL) at 48 h post-incubation (Figure 1E). This result demonstrated the direct bactericidal activity of CoCl_2_ against *Brucella.*

### 2.2. CoCl_2_ Treatment Affected the Survival of B. abortus in RAW 264.7 Macrophage Cells at 4 h pi

The bacterial intracellular growth assay result displayed a significant reduction in bacterial survival at 4, 24, and 48 h pi in CoCl_2_-treated cells compared to the control cells (Figure 2A). To clarify this potential role of CoCl_2_ in macrophage cells at the early stage of infection, we utilized quantitative real-time PCR (qRT-PCR) to analyze the expression of the genes related to phagolysosome fusion, which is considered a central effector in brucellacidal activity [22]. CoCl_2_ treatment elevated the expression of two trafficking regulators, *LAMP1* and *LAMP2*, and lysosomal enzyme *GLA* upon *B. abortus* infection (Figure 2B). Additionally, the results showed that CoCl_2_ treatment increased the expression of the HIF-1α protein at concentrations of 1 and 10 µg/mL (Figure 2C,D). This transcription factor has been associated with a wide range of diseases and host immunity [23].

### 2.3. CoCl_2_ and HIF-1α Affected the Survival of B. abortus in RAW 264.7 Macrophage Cells at 24 h pi

As shown in Figure 2A, CoCl_2_ treatment decreased the intracellular growth of *B. abortus* at all three examined time points. Thus, we next checked how CoCl_2_ induced host immune responses to eliminate *B. abortus* at 24 h pi. Proinflammatory cytokine TNF-α and lysosomal enzyme HEXB effectively contribute to bactericidal activity at this time point of infection [24,25]. Herein, the transcriptional profiles of *TNF-α* and *HEXB* were evaluated using qRT-PCR. CoCl_2_ treatment upregulated the expression of these genes during *B. abortus* infection compared to the untreated control (Figure 3A). Furthermore, similar to the early time point (4 h) of infection, CoCl_2_ stabilized the HIF-1α protein, proven by immunoblotting assay in both concentrations of CoCl_2_-treated cells (Figure 3B,C). Interestingly, even when the cells were not treated with CoCl_2_, the HIF-1α protein was slightly activated upon *B. abortus* infection. This transcription factor was probably involved in the antimicrobial effect of macrophage cells. To clarify this, we knocked down this gene in RAW 264.7 cells by using a siRNA transfection. After 48 h post-transfection with HIF-1α siRNA, the results displayed an effective reduction in the relative mRNA level of this gene compared to the PBS and negative siRNA control (Figure 3D). After an effective silencing, we then checked its effect on the intracellular growth of *B. abortus* in macrophage cells. HIF-1α-silenced cells slightly increased the number of bacteria at 24 h pi compared to both control cells, while there was no difference in the number of bacteria at 4 h pi upon HIF-1α knockdown (Figure 3E).

### 2.4. CoCl_2_ Administration Mediated Host Immunity against B. abortus Infection in Mice

Following promising results in in vitro experiments, it is necessary to further clarify the role of CoCl_2_ in an in vivo examination using a mouse model in the context of *B. abortus* infection. Mice were administered with two concentrations of CoCl_2_ (20 and 40 mg/kg/day). On day 14 after infection, the serum cytokines measurement was processed for evaluating the host immune responses. Cytokines IFN-γ and IL-6 play essential roles in killing *Brucella*. Our results showed a significant increase in the production of IFN-γ and IL-6 in higher concentrations of the CoCl_2_-treated group (Figure 4A,B). Moreover, both treatment groups displayed a lower bacterial load in the spleen and the spleen weight than the control group (Figure 4C,D). CoCl_2_ treatment at a high concentration also decreased the bacterial load in the liver and liver weight compared to the control group. These results indicated the effect of CoCl_2_ on mediating the host immunity and protecting against *B. abortus* infection.

## 3. Discussion

Macrophage is known as the first line of innate immunity. An outstanding function of macrophages is phagocytosis, by which they can ingest foreign particles or pathogenic microorganisms. *Brucella* are intracellular bacteria, and the invasion into macrophages is considered a critical determinant of their intracellular growth fate [26]. This is initiated by the interaction of *Brucella* with macrophage membranes through lipid rafts via TLR4. Furthermore, the intracellular circulation of *Brucella* has entirely fulfilled in autophagic *Brucella*-containing vacuoles as the end point where it can be released to continue infecting other host cells [27]. This current study showed that CoCl_2_ treatment reduced the phagocytosis of *B. abortus* into RAW 264.7 macrophage cells. To investigate how CoCl_2_ could mediate this event, we evaluated the expression of regulators related to macrophage phagocytosis, including F-actin and ERK1/2 MAPK. The TLR4/ERK1/2 MAPK/F-actin axis is necessary for *Brucella* internalization and has been exploited in many previous studies, even in the infection of other bacteria [28,29,30]. A study done by Murata et al. [31] displayed the impact of CoCl_2_ treatment on the disruption of F-actin in both HepG2 and PLC/PRF/5 cell lines. Similarly, our data showed a reduction in F-actin expression and the phosphorylation of ERK1/2 at 30 min pi. Notably, although CoCl_2_ stabilizes the transcription factor HIF-1α in normoxic conditions, we found that the expression of HIF-1α was decreased in CoCl_2_-treated cells during *Brucella* infection at the same examined time point. HIF-1α is linked to macrophage phagocytosis via p38 MAPK activation and the activated TLR4/ERK1/2 MAPK/NF-ĸB pathway [32,33]. In addition, a study by Shweta et al. [34] showed that CoCl_2_ treatment reduced the expression of TLR4, which is the crucial receptor for *Brucella* invasion into macrophages. Therefore, our results suggested that CoCl_2_ reduced *Brucella* uptake through the MAPK signaling pathway, which is probably related to the stabilization of HIF-1α. A comprehensive evaluation using gene silencing or other molecular methods would provide further insight into the probable role of HIF-1α in *Brucella* invasion into macrophages.

After successful internalization, the phagosomes containing *Brucella* fuse with lysosomes. The phagolysosome fusion is one of the macrophage brucellacidal activities that contributes to a 90% reduction of *Brucella* viability at the early stage of infection. However, this is a struggle for survival; hence, *Brucella* has acquired crafty evasive strategies to prevent this event that facilitates its intracellular growth [35]. The T4SS system of *Brucella* successfully accomplishes its role as a critical virulence factor by releasing a variety of effectors that facilitate trafficking and arrival at the endoplasmic reticulum and prevents the host killing mechanism by phagolysosome fusion [36]. Strikingly, CoCl_2_ treatment decreased the intracellular growth of *B. abortus* at all examined time points. As mentioned above, phagolysosome fusion plays a vital role in *Brucella* elimination. Trafficking regulators LAMP1 and LAMP2, as well as lysosomal enzymes GLA and HEXB, are critical markers for this event [25,37]. In addition to these potential weapons, macrophages can secrete cytokines, especially TNF-α, to regulate various bactericidal effectors [24]. For this reason, we sought to evaluate their expression in CoCl_2_ treatment during *B. abortus* infection. Our results displayed an increase in the transcripts of *GLA*, *LAMP1*, and *LAMP2* at an early time point, as well as *HEXB* and *TNF-α* at 24 h pi. These findings agree with the previous studies that showed an enhancement of LAMP1 colocalization in HepG2 cells and TNF-α production in BV2 cells [38,39]. As proven earlier, HIF-1α was considered to relate to macrophage phagocytosis. Thus, we next elucidate the involvement of HIF-1α expression in *B. abortus* intracellular growth. HIF-1α has emerged as a potential key to improving host immune responses against intracellular bacterial, fungal, and protozoan pathogens. In particular, HIF-1α-deficient mice enhanced the survival of *Listeria monocytogenes* (*L. monocytogenes*) and *Mycobacterium avium* (*M. avium*) in the liver [40]. Moreover, a study done by Li et al. [41] showed that HIF-1α promoted a macrophage inflammatory response by producing cytokine TNF-α during *L. monocytogenes* infection, and CoCl_2_ treatment stabilized HIF-1α expression, as well as induced TNF-α production, upon *Candida albicans* yeast infection. This is consistent with our results of the CoCl_2_ treatment at 24 h pi. In addition, HIF-1α silencing slightly increased *B. abortus* survival at this tested time point. On the other hand, our data about the expression of the HIF-1α protein is notable. CoCl_2_ treatment increased the HIF-1α proteins at 4 and 24 h during *B. abortus* infection. CoCl_2_ is well-known to stabilize HIF-1α at the post-translational level by inhibiting the propyl hydroxylase, which is mainly responsible for HIF-1α degradation in the presence of oxygen [42]. CoCl_2_ was also proven to enhance the NF-ĸB activity, leading to increasing the expression of proinflammatory mediators such as IL-6 and iNOS, which are beneficial in controlling *Brucella abortus* [43]. Moreover, NF-ĸB was demonstrated to be a direct modulator of HIF-1α transcription [44]. These results strengthened the role of CoCl_2_ and the distribution of HIF-1α in the macrophage defense against intracellular bacterial infection. This effect of CoCl_2_ is probably related to the transcription factor NF-ĸB.

On the other hand, Cobalt (II) has been considered a potential ligand to form complexes with that have antimicrobial properties. The novel Cobalt (II) complexes with various antibiotics displayed an extensive range of antimicrobial activity against many pathogens [45,46]. Moreover, CoCl_2_ was examined directly as a specific antimicrobial agent against *H. pylori* through competition with nickel ions [10]. Likewise, we first reported that CoCl_2_ displayed a direct inhibitory effect against *B. abortus.* This is possibly due to the inhibition of the synthesis of cyanide-sensitive oxidative enzymes [47]. This result raised the question of whether CoCl_2_ could be a potential alternative strategy to control animal brucellosis. Together with obtaining the other promising in vitro results, we next treated mice with CoCl_2_ at concentrations of 20 and 40 mg/kg/day, demonstrating the acquisition of a hypoxia-like condition [48]. CoCl_2_ was involved in the induction of interferon and IL-6 production. More evidence has proposed a functional role in activating the host immunity of proinflammatory cytokines IFN-γ and IL-6 in the control of *Brucella* infection [49,50,51,52]. In this study, CoCl_2_ administration in mice elevated the production of IFN-γ and IL-6, leading to reducing the bacterial load in the spleen and liver. In addition, splenomegaly and hepatomegaly observed in *Brucella*-infected mice were reduced in CoCl_2_-treated mice. Moreover, HIF-1α has been implicated in the protection against *M. avium* and *L. monocytogenes* in a mouse model through the activation of immune effectors, which is beneficial in controlling intracellular bacteria [40,41].

## 4. Materials and Methods

### 4.1. Cell Culture and Bacterial Growth Condition

Murine macrophage RAW 264.7 cells (ATCC, TIB-71) were cultured at 37 °C in 5% CO_2_ atmosphere in RPMI 1640 medium (Gibco, 11875119) containing 10% (*v*/*v*) heat-inactivated fetal bovine serum (FBS) (Gibco, 1600-044) with or without 1% antibiotics (Gibco, 15140122). For the following assays, the cells were seeded at concentrations of 3 × 10^4^ and 6 × 10^5^ cells per well in 96-well and 6-well cell culture plates, respectively, in a culture medium containing RPMI plus 10% FBS. The smooth, virulent, wild-type *B. abortus* 544 biovar 1 strain (ATCC 23448) was cultured in Brucella broth (BBL BD, USA) at 37 °C until the stationary phase or Brucella broth containing 1.5% agar, followed by three days incubation for colony-forming unit (CFU) counting.

### 4.2. CoCl_2_ Solution Preparation and Cell Viability Assessment Assay

CoCl_2_ (Sigma, C8661) was dissolved in distilled water (DW) to make a concentration of 10 mg/mL as a stock solution. The stock solution was sterilized by filtration using a 0.22-µm-pore size membrane.

Cell viability assessment was performed using the MTT assay. Cells were subcultured in a 96-well plate. After overnight incubation, cells were treated with different concentrations of CoCl_2_ (10 mg/mL, 1 mg/mL, 100 µg/mL, 10 µg/mL, 1 µg/mL, 100 ng/mL, and 10 ng/mL) in 100 µL of RPMI plus 10% FBS for 72 h. Afterward, the medium was changed to a new medium containing 5 mg/mL of MTT solution (Sigma, M5655) and incubated for 4 h. DMSO was added to each well to dissolve the formazan crystals, then measured the absorbance at an optical density of 540 nm using a spectrophotometer (Thermo Labsystems Multiskan, Chantilly, VA, USA).

### 4.3. Bacterial Invasion and Intracellular Growth Assay

For the invasion assay, after the overnight culture, the cells were treated with 10 µg/mL of CoCl_2_ for 6 h, while the control cells were treated with DW. Treated cells were infected with *B. abortus* at a multiplicity of infection of 20 by centrifuging at 150× *g* for 10 min at room temperature (RT). The infected cells were then incubated for 15, 30, and 45 min at 37 °C. At each time point, the culture medium was changed to a new medium containing 50 µg/mL of gentamicin (Gibco, 15710-064) and incubated for 30 min to kill the extracellular bacteria. The cells were then washed twice with PBS and lysed with DW, followed by serial dilution, and plated on an agar plate to enumerate the number of invaded bacteria.

The same procedure was performed as that of the invasion assay for the intracellular replication assay, with some modifications. Briefly, at one hour pi, the cells were treated the same as the pretreatment protocol in a new medium containing 50 µg/mL of gentamicin and incubated for 4, 24, and 48 h. At the indicated time point, the washing, lysing, and plating procedures were done the same way as the invasion assay. After three days of incubation, the number of CFU was counted, and a base-10 logarithm was calculated.

### 4.4. Western Blot

The immunoblotting analysis was performed as previously described [53]. Cells were subcultured in a 6-well plate and incubated overnight. After that, the cells were treated as mentioned above in the invasion and intracellular growth assays. At the indicated time points, the cells were lysed in 200 µL of RIPA lysis buffer (Pierce, 89900) containing 1% of protease inhibitor cocktail (Promega, G6521, Madison, WI, USA) to collect the total cellular proteins, followed by BCA protein quantification. The protein samples were subjected to SDS-PAGE and then transferred onto the immobilon-P membrane (Millipore, Burlington, MA, USA). The membrane was blocked with blocking buffer (5% skim milk (BD, 232100) in TBS-Tween 20) for 30 min at RT. Afterward, the membrane was incubated overnight at 4 °C with different primary antibodies that were diluted in blocking buffer, including F-actin (1:400; Bioss, BS-1571R, Woburn, MA, USA), phospho-ERK1/2 (1:500; Cell signaling, 4377S, Danvers, MA, USA), pan-ERK1/2 (1:500; Cell signaling, 4695S), HIF-1α (1:2000; Invitrogen, Carlsbad, CA, USA, PA1-16601 or 1:1000; Cell signaling, 14179S), and β-actin (1:2000; Cell signaling, 4967S). Following binding with the primary antibodies, the membrane was washed three times with TBS-Tween 20 for 20 min and then incubated with a secondary antibody (1:2000; Cell signaling, 7074S). Finally, the membrane was washed three times with the same washing buffer for 10 min, and the protein bands were visualized using EzWestLumi chemiluminescent substrate (Atto, WSE-7120L, Tokyo, Japan).

### 4.5. Bactericidal Effect Assay

The CoCl_2_ stock solution was diluted by using RPMI medium and added to 96-well plate to reach various concentrations (1 µg/mL, 10 µg/mL, 100 µg/mL, and 1000 µg/mL). After that, *Brucella* at a density of 1 × 10^4^ CFU was added to each well containing different concentrations of CoCl_2_ and 90 μL of PBS. The bactericidal effect of CoCl_2_ was evaluated at 37 °C for 0, 6, 24, and 48 h. At specific time points, each diluent was plated onto Brucella agar and cultured for three days to determine the CFUs.

### 4.6. Ribonucleic Acid (RNA) Isolation and Quantitative Real-Time PCR

To investigate the expression of genes that related to the host immune responses against *B. abortus* infection, the overnight cultured macrophage cells in a 6-well plate were treated as mentioned in the bacterial intracellular growth assay. At the indicated time points, the total RNA was extracted using a RNeasy mini kit (Qiagen, 74104, Hilden, Germany). The RNA concentration was measured and equalized for the complementary DNA (cDNA) synthesis, which was performed by the QuantiTech Reverse Transcription Kit (Qiagen, 205311). The synthesized cDNA was used as the template for SYBR Green-based qRT-PCR (Promega, A6002). The thermal cycle conditions were 95 °C for 10 min, 39 cycles with 95 °C for 15 s and 60 °C for 1 min, followed by a melting curve analysis from 65 °C to 95 °C with an increment of 1 °C each 5 s using the CFX96 Touch Real-Time PCR Detection System (Bio-Rad, Contra Costa County, CA, USA). The results were analyzed with Bio-Rad CFX Manager software, version 3.1, and the relative fold change of the mRNA level was calculated using the 2^−^^∆∆^^CT^ method. All kits were used according to the manufacturer’s protocol, and the primers for all the related genes are listed in Table 1.

### 4.7. siRNA Knockdown

Cells were transfected with HIF-1α siRNA (Santa Cruz, sc-35562), as previously reported [54]. Briefly, cells were grown at concentrations of 4 × 10^5^ cells per well in a 6-well plate and 10^4^ cells per well in a 96-well plate for cDNA template preparation and the bacterial intracellular survival assay, respectively. After that, siRNA transfection into the cells was accomplished by using Lipofectamine RNAiMAX (Invitrogen, 13778150) with 60 pmoL per well in a 6-well plate and 3 pmoL per well in a 96-well plate. At 48 h post-transfection, qRT-PCR was utilized to assess the silencing efficiency. The intracellular growth assay mentioned above was performed to evaluate the role of HIF-1α in *B. abortus* infection. The negative siRNA control (Santa Cruz, sc-37007) was used at the concentrations of 60 pmoL per well in a 6-well plate and 2 pmoL per well in a 96-well plate. 

### 4.8. Mice Treatment with CoCl_2_ and Protection Experiment

Fifteen twelve-week-old female B6 mice (Samtako, Korea) were first acclimatized for one week and then distributed into three groups of five mice each. Two groups were orally pretreated with 20 mg/kg/day and 40 mg/kg/day of CoCl_2_, while the control group was given with DW in a total volume of 100 µL for three days prior to infection. All mice were then intraperitoneally (IP) infected with 2 × 10^5^ CFUs of *Brucella* in 100 µL of PBS per mouse and started on an additional 14-day treatment regimen. After that, all mice were sacrificed by cervical dislocation, and the spleens and livers were aseptically removed. The bacterial load in 0.05 g of homogenized spleen and liver was determined by serial dilution on a Brucella agar plate. In addition, the spleen and liver weights were measured to evaluate the splenomegaly and hepatomegaly. Finally, a base-10 logarithm of the number of CFU was calculated to evaluate the protective effect of CoCl_2_ against *Brucella* infection in vivo.

### 4.9. Serum Cytokine Level Measurement

The level of IFN-γ and IL-6 cytokines in serum reflects the host immune responses against *Brucella* infection. Hence, the peripheral blood samples were collected via tail vein at two weeks pi, followed by centrifugation at 2000× *g* at 4 °C for 10 min, to collect the serum samples. After that, the concentration of these cytokines in the serum samples was determined by Cytometric Bead Array (BD CBA Mouse Inflammation Kit, 552364) and analyzed using BD FACSVerse flow cytometry.

### 4.10. Statistical Analysis

The data were expressed as the means ± the standard deviations (SD). Statistical analysis was performed with GraphPad InStat using an unpaired Student’s *t-*test. The results with *, *p* < 0.05; **, *p* < 0.01; and ***, *p* < 0.001 were considered statistically significant.

## 5. Conclusions

In conclusion, our study contributes to a better understanding of CoCl_2_ effects on host immunity against *B. abortus* infection. In particular, CoCl_2_ treatment reduced the phagocytosis of *B. abortus* into RAW 264.7 macrophage cells, possibly through the F-actin/ERK1/2 MAPK signaling pathway. In controlling the intracellular replication of *B. abortus*, CoCl_2_ treatment activated effectors that promoted host immune responses both in vitro and in vivo. Meanwhile, CoCl_2_ is well-known to not only stabilize HIF-1α but also activate the NF-ĸB signaling pathway. Therefore, further, deeper investigations should be performed to clarify the reciprocal regulation of the pivotal transcription factors HIF-1α and NF-ĸB in the context of the invasion and intracellular growth of *B. abortus* in a cell line or animal model.

## Figures and Tables

**Figure 1 pathogens-11-00596-f001:**
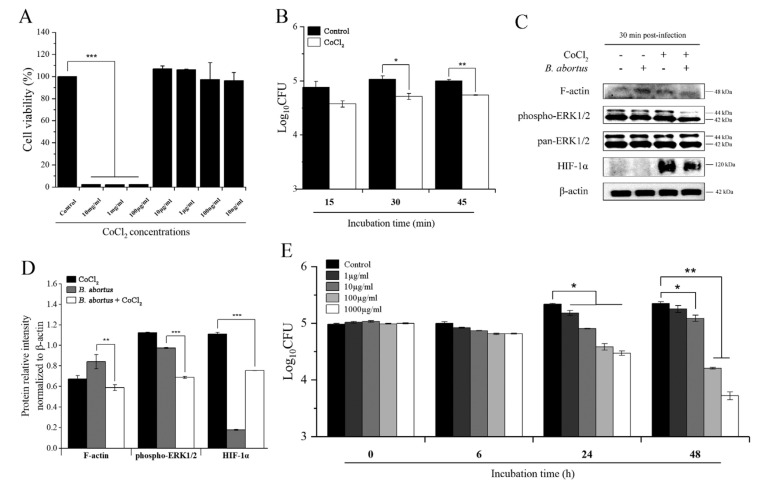
The effect of CoCl_2_ on RAW 264.7 cells and *B. abortus* viability and *B. abortus* internalization into macrophage cells. RAW 264.7 cells were pretreated with different concentrations of CoCl_2_ for 72 h, and the cell viability was evaluated using the MTT assay (**A**). Cells were pretreated with 10 µg/mL of CoCl_2_ for 6 h, and the number of invaded *Brucella* was determined at 15, 30, and 45 min pi (**B**). The involvement of F-actin, ERK1/2, and HIF-1α proteins in the phagocytosis signaling pathway at 30 min pi was determined using an immunoblotting assay (**C**). Protein intensity from the Western blot bands was analyzed by ImageJ software and normalized relative to β-actin (**D**). The direct bactericidal effect of CoCl_2_ on *Brucella* survival was evaluated for 0, 6, 24, and 48 h (**E**). The data are represented as the mean ± SD of duplicate samples from at least two independent experiments. Statistically significant differences relative to the control group are indicated by an asterisk (* *p* < 0.05, ** *p* < 0.01, and *** *p* < 0.001).

**Figure 2 pathogens-11-00596-f002:**
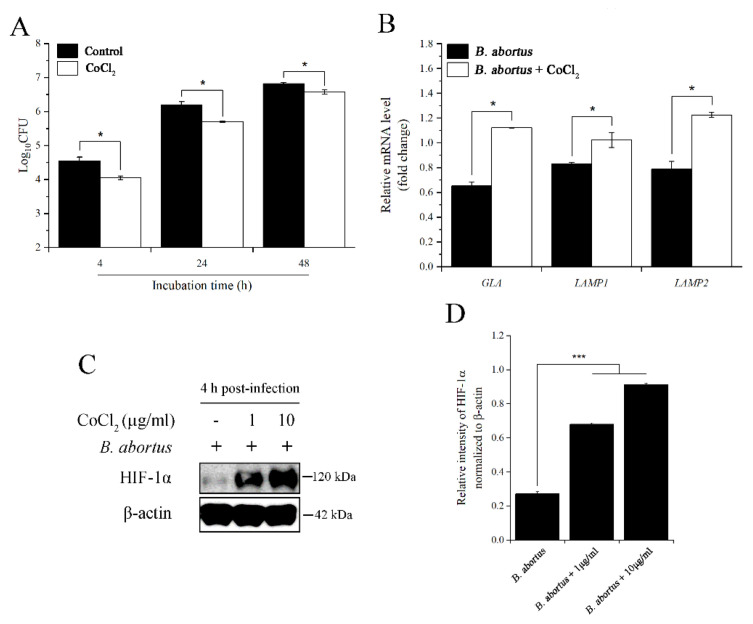
The effects of CoCl_2_ on the intracellular growth of *B. abortus* at 4 h pi. Cells were treated with 10 µg/mL of CoCl_2_, and the number of intracellular bacteria was determined at 4, 24, and 48 h pi (**A**). At 4 h pi, RNA extraction and cDNA synthesis were performed, followed by qRT-PCR, to check the transcripts of *GLA*, *LAMP1*, and *LAMP2* genes (**B**). At the same time, the immunoblotting assay was used to check the expression of the HIF-1α protein using two concentrations of 1 and 10 µg/mL of CoCl_2_ (**C**), and the relative protein intensity normalized to β-actin was carried out by ImageJ software (**D**). The data are represented as the mean ± SD of duplicate samples from at least two independent experiments. Statistically significant differences relative to the control group are indicated by an asterisk (* *p* < 0.05 and *** *p* < 0.001).

**Figure 3 pathogens-11-00596-f003:**
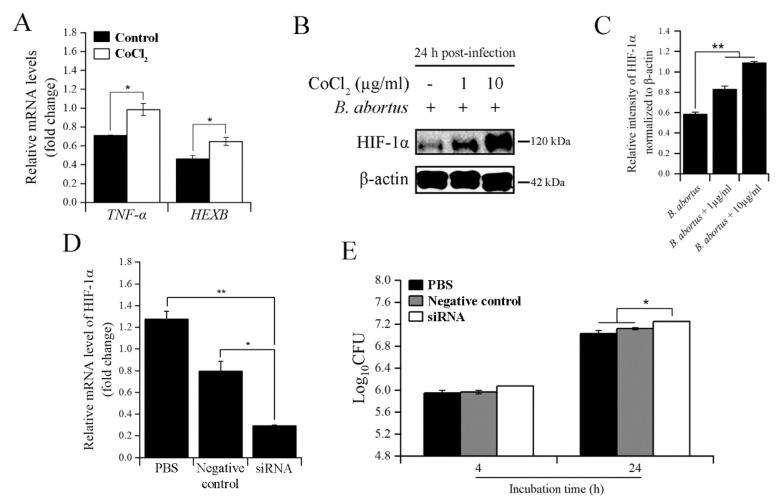
The effects of CoCl_2_ and HIF-1α on the intracellular growth of *B. abortus* at 24 h pi. The transcripts of *TNF-α* and *HEXB* were determined using qRT-PCR at 24 h pi (**A**). At the same time, the immunoblotting assay was used to check the expression of the HIF-1α protein using two concentrations of CoCl_2_ (1 and 10 µg/mL) (**B**), and the relative protein intensity normalized to β-actin was carried out by ImageJ software (**C**). Cells were transfected with HIF-1α siRNA for 48 h, and qRT-PCR was utilized to evaluate the transfection efficacy (**D**). The successful knockdown of the HIF-1α gene was used to determine the bacterial intracellular growth at 4 and 24 h pi (**E**). The data are represented as the mean ± SD of duplicate samples from at least two independent experiments. Statistically significant differences relative to the control group are indicated by an asterisk (* *p* < 0.05 and ** *p* < 0.01).

**Figure 4 pathogens-11-00596-f004:**
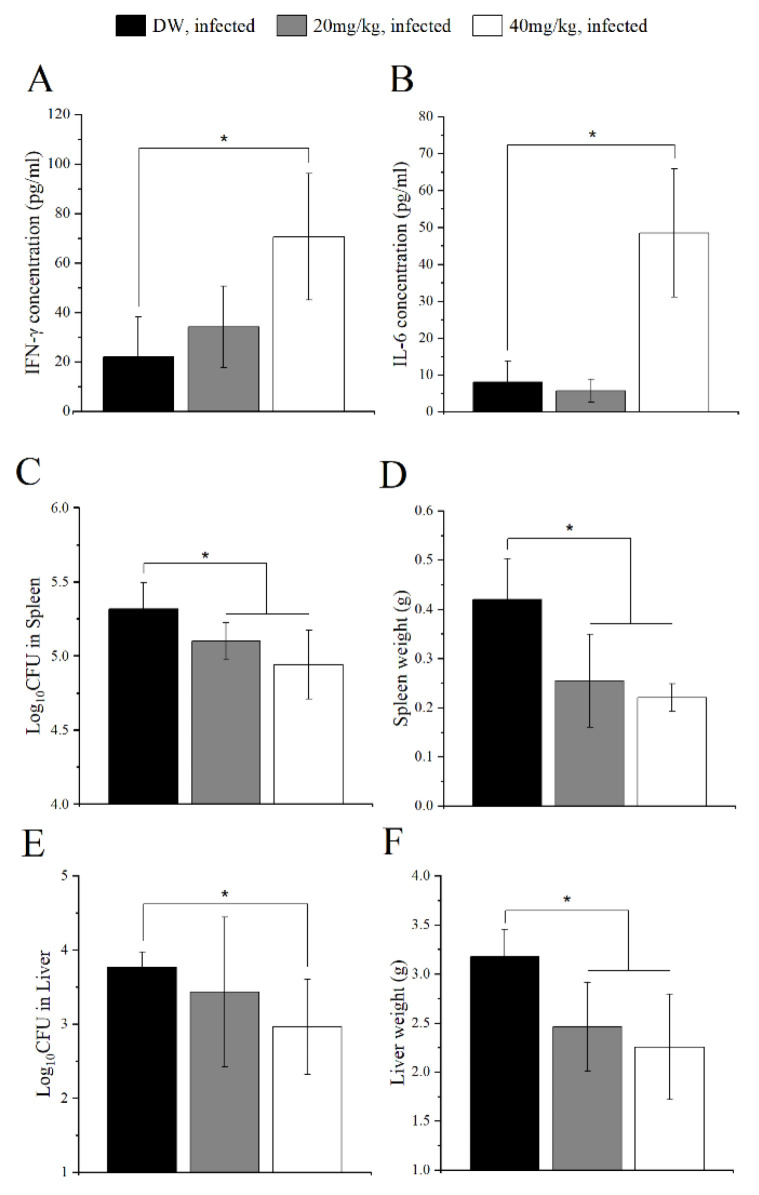
Protection against *B. abortus* in B6 mice treated with CoCl_2_. B6 mice were orally administered with 20 or 40 mg/kg/day of CoCl_2_ for three days prior to infection. After that, the mice were IP challenged with *B. abortus*, followed by a continuous 14-day treatment regimen. At day 14 pi, the serum was collected to evaluate the production of cytokines IFN-γ (**A**) and IL-6 (**B**). At the same time, mice were sacrificed, the spleens and livers were collected and homogenized, and the number of CFU in each spleen and liver was counted (**C**,**E**). The weights of the spleen and liver were evaluated. (**D**,**F**) The data are represented as the mean ± SD of each group of five mice. Asterisks indicate statistically significant differences (* *p* < 0.05).

**Table 1 pathogens-11-00596-t001:** Primer sequences used for qRT-PCR.

**Gene**	**Description**	**Forward Primer**	**Reverse Primer**
GLA	Galactosidase, alpha	5′-GGC CAT GAA GCT TTT GAG CAG AGA TAC-3′	5′-AGT CAA GGT TGC ACA TGA AAC GTT CCC-3′
LAMP1	Lysosomal-associated membrane protein 1	5′-GGC CGC TGC TCC TGC TGC TGC TGG CAG-3′	5′-ATA TCC TCT TCC AAA AGT AAT TGT GAG-3′
LAMP2	Lysosomal-associated membrane protein 2	5′-AGG GTA CTT GCC TTT ATG CAG AAT GGG-3′	5′-GTG TCG CCT TGT CAG GTA CTG AAT GG-3′
HIF-1α	Hypoxia inducible factor 1, alpha subunit	5′-TCC CAT ACA AGG CAG CAG AA-3′	5′-GTG CAG TAT TGT AGC CAG GC-3′
TNF-α	Tumor necrosis factor, alpha	5′-CAG GTT CTG TCC CTT TCA CTC ACT-3′	5′-GTT CAG TAG ACA GAA GAG CGT GGT-3′
HEXB	Hexosaminidase B	5′-CCC GGG CTG CTG CTG CTG CAG GCG CTG-3′	5′-GTG GAA TTG GGA CTG TGG TCG ATG CTG-3′
β-ACTIN	Actin, beta	5′-CGC CAC CAG TTC GCC ATG GA-3′	5′- TAC AGC CC GGG GAG CAT CGT-3′

## Data Availability

The data described in this study are available for sharing.

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
