# Peer review of "Cobalt (II) Chloride Regulates the Invasion and Survival of Brucella abortus 544 in RAW 264.7 Cells and B6 Mice"

_pathogens, 2022, doi:10.3390/pathogens11050596_

Round 1
Reviewer 1 Report
The authors have presented a well prepared manuscript on the ability of CoCL2 to stimulate a protective immune response to B. abortus. The manuscript is well written and has a compelling presentation of data from well designed experiments. No major concerns or faults were noted. However, a few minor issues were identified. The readability of the manuscript will be improved if the authors can address the following issues:
- In the results section and Figure 1E. the author's should clarify if the data represent the direct effects of CoCl2 on B. abortus only. This condition was assumed but some readers may interpret these data as coming from infected RAW cell culture.
- The statistical analysis section of the methods (line 355-357) only describes the use of a Student's T test for data comparisons. While this method would be appropriate for data comparisons between two groups it may be problematic for multi-group comparisons and may not define correct levels of statistical significance. The author's should either expand and correctly label the statistical methods used for each data analysis or add the correct statistical method to the data analysis and the methods section.
Author Response
May, 2022
Dear Dr. Tessie Mi,
Editor of Pathogens Journal
Enclosed hereby, please find our revised manuscript (ID: pathogens-1709434) entitled: “Cobalt (II) chloride regulates the invasion and survival of Brucella abortus 544 in RAW 264.7 cells and B6 mice”.
We would like to thank the Editors and the reviewers for their constructive comments which we took into consideration when revising our manuscript. All of the comments raised by the reviewers have been addressed in detail in our efforts to improve our manuscript and all the changes that we made in response to the reviewers’ comments are highlighted in the yellow text in the revised manuscript. A point-by-point response to the reviewers’ comments follows on the accompanying pages.
We hope that the revised manuscript is now acceptable for publication in Pathogens Journal. Please contact me with any questions concerning the manuscript. I can be reached at +82-55-772-2359 or by e-mail at kimsuk@gnu.ac.kr.
With regards,
Suk Kim, Ph. D.
College of Veterinary Medicine,
Gyeongsang National University,
Jinju, 660-701, Republic of Korea

Reviewer 2 Report
This manuscript describes the effect of cobalt (II) chloride (CoCl2) on an interaction of Brucella abortus (B. abortus) with RAW264.7 cells by focusing on HIF-1a and that on mice infected with B. abortus. Although this study was well designed and well written, there are some flaws in the manuscript.
(1) The effect of CoCl2 on an interaction of B. abortus with RAW264.7 cells appears to be nicely correlated with the level of HIF-1a. However, knock down of HIF-1a caused a significant but slight reduction of intracellular B. abortus, indicating that HIF-1a may not be a sole key molecule for intracellular growth of B. abortus. If the authors believe so, they should state it clearly. In relation to this, the authors stated in conclusion that CoCl2 is well-known to not only stabilize HIF-1a but also activate the NF-kB signaling pathway. If the authors believe that this may be the possible reason for the results of knock down of HIF-1a, then they should clearly state it.
(2) Intracellular bacteria may be killed by cell penetrating antibiotics such as quinolone and macrolide. The authors should discuss about the drawbacks of such antibiotics upon application to B. abortus. In addition, the authors should discuss about the possibility that CoCl2 can be employed as an alternative.
In each Figure, panels should be placed from left to right in each row.
Author Response

(The authors gave the same response as above.)

Reviewer 3 Report
CoCl2 inhibits the invasion and intracellular survival of Brucella abortus on RAW 264.7 macrophages and mice. CoCl2 helps to improve the host's immunomodulatory effect on the infection of Brucella abortus, which may provide a new idea for the prevention and treatment of brucellosis. However, there are several small problems that need to be solved.
- Brucella is a facultative intracellular parasite, which should be reflected in the discussion part of the article. In addition, these two literatures can be cited:
Xiong X, Li B, Zhou Z, Gu G, Li M, Liu J, Jiao H. The VirB System Plays a Crucial Role in Brucella Intracellular Infection. Int J Mol Sci. 2021 Dec 20;22(24):13637. doi: 10.3390/ijms222413637. PMID: 34948430; PMCID: PMC8707931.
Jiao H, Zhou Z, Li B, Xiao Y, Li M, Zeng H, Guo X, Gu G. The Mechanism of Facultative Intracellular Parasitism of Brucella. Int J Mol Sci. 2021 Apr 1;22(7):3673. doi: 10.3390/ijms22073673. PMID: 33916050; PMCID: PMC8036852.
- Please provide the qualification certificate of animal experiment and animal ethics certificate.
- RAW264.7 cells are difficult to transfect. Has the transfection efficiency been tested? In addition, the concentration of siRNA is pmol. Have you explored the best concentration to optimize the silencing effect.
- Please provide the original uncut picture of WB.
Author Response

(The authors gave the same response as above.)

Round 2
Reviewer 2 Report
In all figures, panels should be placed from left to right in each row.
Author Response
Q1: In all figures, panels should be placed from left to right in each row.
Authors answer to Q1: As per reviewer’s suggestion, we revised all figures as per reviewer’s comment.
